# miRNAs in Lymphocytic Leukaemias—The miRror of Drug Resistance

**DOI:** 10.3390/ijms23094657

**Published:** 2022-04-22

**Authors:** Yordan Sbirkov, Bozhidar Vergov, Nikolay Mehterov, Victoria Sarafian

**Affiliations:** 1Department of Medical Biology, Medical University of Plovdiv, 4002 Plovdiv, Bulgaria; bozhidar.vergov@mu-plovdiv.bg (B.V.); nikolay.mehterov@mu-plovdiv.bg (N.M.); 2Division of Molecular and Regenerative Medicine, Research Institute at Medical University of Plovdiv, 4002 Plovdiv, Bulgaria

**Keywords:** miRNA, drug resistance, lymphocytic leukaemia

## Abstract

Refractory disease and relapse remain the main causes of cancer therapy failure. Refined risk stratification, treatment regimens and improved early diagnosis and detection of minimal residual disease have increased cure rates in malignancies like childhood acute lymphoblastic leukaemia (ALL) to 90%. Nevertheless, overall survival in the context of drug resistance remains poor. The regulatory role of micro RNAs (miRNAs) in cell differentiation, homeostasis and tumorigenesis has been under extensive investigation in different cancers. There is accumulating data demonstrating the significance of miRNAs for therapy outcomes in lymphoid malignancies and some direct demonstrations of the interplay between these small molecules and drug response. Here, we summarise miRNAs’ impact on chemotherapy resistance in adult and paediatric ALL and chronic lymphocytic leukaemia (CLL). The main focus of this review is on the modulation of particular signaling pathways like PI3K-AKT, transcription factors such as NF-κB, and apoptotic mediators, all of which are bona fide and pivotal elements orchestrating the survival of malignant lymphocytic cells. Finally, we discuss the attractive strategy of using mimics, antimiRs and other molecular approaches pointing at miRNAs as promising therapeutic targets. Such novel strategies to circumvent ALL and CLL resistance networks may potentially improve patients’ responses and survival rates.

## 1. Introduction

Cell function and fate are ultimately determined by the type and amount of proteins produced in the cell and their post-translational modifications. Control over protein synthesis is exerted at multiple levels through epigenetics, transcription and mRNA export regulation, stability, post-transcriptional modifications, fine-tuning of the translational machinery, including ribosomes, initiation factors, etc. Importantly, a complex system at the interface between transcription and translation further orchestrates the cell’s proteome–micro RNA (miRNAs) dynamics. Ranging from merely ~20 to 25 nucleotides in length, miRNAs can be transcribed from both intra- and inter-genic regions, after which they are processed in a tightly regulated step-wise manner to ultimately control mRNA degradation, decay or even to determine rates of translation [1]. The mechanisms of miRNA biogenesis and control of translation (and even of transcription through promotor binding) are remarkably versatile and have been extensively reviewed [1,2].

Currently, there are more than 2500 confirmed miRNAs in the human genome, and it is estimated that they may regulate at least one-third of our mRNAs (or even up to 15,000 genes) [3,4,5]. The picture’s complexity is further rendered because tens of miRNAs may target one gene, and one miRNA could have tens to hundreds of mRNA clients [5,6]. Of note, miRNAs can be specifically targeted by cancer cells, as it has been shown that a large proportion of these short regulatory oligonucleotides are encoded in fragile sites in the genome that are frequently deleted or silenced [7]. Examples of such genomic perturbation in leukaemias include miR-15a/16-1 deletion as part of chromosome 13q14, miR-125a translocation to *IGH*, and others (discussed later) [7,8,9].

Acute and chronic lymphocytic leukaemias (ALL and CLL) are the most common haematological malignancies in children and adults. ALL is a heterogeneous disease arising from haematopoietic blast cells characterised by incomplete differentiation, loss of function, and mutations. Interestingly, ALL has a bimodal distribution, with one peak at age 5 (80% of ALL cases) and another at 50 years of age [10]. Childhood ALL (cALL) is treated successfully, leading to ~90% cure rates. In contrast, albeit rare (<1% of all cancers), the outcome in adult ALL remains poor even if therapy is closely related in both groups [11,12]. Similarly, CLL is a disease of the elderly (median age of ~70) characterised by the expansion of slowly proliferating CD5/CD19 positive B cell clones. There are a number of different cytogenetic modalities influencing the progression of the disease and the treatment protocols. Therefore, the 5-year overall survival (OS) can range from 90% down to merely ~10% in high-risk groups with deletions of *TP53* [13,14]. While these blood cancers have different aetiologies, cytogenetics and dynamics, they share certain similarities, especially in the treatment strategies. Resistance to the commonly used purine analogues, alkylating agents, anthracyclines and glucocorticoids are the main reasons for drug failure. Even though the 5-year overall survival (OS) can reach 90% for childhood ALL and CLL [15,16], this percentage is only around 60% for younger adults with ALL and just ~10% for older patients with ALL [17]. Thus, the lack of therapy response can turn these well-manageable diseases into daunting battles to fight. Therefore, it is urgent to study the underlying reasons for drug resistance and the potential strategies to overcome them.

The role of miRNAs in regulating normal haematopoiesis, leukaemogenesis and as diagnostic and prognostic markers in cancer has been under extensive investigation over the last decade. Thus, several studies have elucidated a plethora of functions of these small RNAs in blood cancers [18,19,20,21,22]. For instance, a specific miRNA signature correlates with prognosis and disease progression in CLL [23], or subsets of miRNAs can be used to differentiate between childhood B cell and T cell ALL [22]. Klein et al. have demonstrated that miR-15a/16-1, frequently deleted (in ~60%) in CLL patients, acts as tumour suppressors by controlling cell cycle progression. Furthermore, the deletion of these miRNAs can even drive CLL in a mouse model, while their ectopic expression can reduce cell growth in both mouse and human cells [24]. Similarly, in in vivo models, the overexpression of miR-125b (by retroviruses or by placing the miRNA after an *IGH* enhancer and promoter) can induce B cell lymphoblastic leukaemia proving the direct role of miRNAs in leukaemogenesis [25,26].

The role of miRNAs in chemoresistance in solid tumours has been well-established [27,28]. Therefore, here we summarise current knowledge of the link between certain miRNAs and molecules regulating classical cell growth, renewal and survival signaling pathways or directly modulating the cell cycle, apoptosis and other important processes related to chemoresistance in acute and chronic lymphocytic leukemias. We finally discuss the perspectives of implementing miRNAs as therapeutic targets in the clinic.

## 2. Implication of miRNAs in Drug Resistance in Adult ALL

ALL in adults accounts for ~0.3% of all malignancies [12]. About 75% of cases arise from B cells, while the remaining 25% of patients present with T cell ALL. Mortality rates remain high, with merely 30–40% achieving complete remission. Around 50% of patients eventually relapse, and there is only 10% 5-year survival with refractory/relapse disease [29]. Such poor treatment outcomes may be due to age-related intolerance to intensified therapy, the accumulation of mutations and higher in vitro drug resistance compared to childhood ALL, where cure rates reach 90% [30]. Adult ALL shows great cytogenetic heterogeneity, with subsets harbouring translocations like t(4;11), t(1;19), t(12;21) or t(9;22) resulting in MLL-AF4, PBX-E2A, ETV6-AML1, or BCR-ABL fusion proteins, respectively [31]. B cell ALL with BCR-ABL translocations or with Philadelphia chromosome (Ph)-like gene expression patterns are the two most common sub-types characterising almost half of cases (15–25% and up to 30% respectively) and showing the worst outcome [10,32]. Standard chemotherapy in adult ALL, similarly to childhood ALL, relies on glucocorticoids (GC) (prednisolone or dexamethasone), anthracyclins (daunorubicin, doxorubicin, idarubicin), cyclophosphamide or cytarabine. Treatment could include L-Asparaginase and targeted therapies based on antibodies (monoclonal or bi-specific) and tyrosine kinase inhibitors for Ph+ patients [31]. Resistance to some of the abovementioned and other drugs and a decrease in apoptosis are events connected to the dysregulation of certain miRNAs (described below and summarised in Table 1).

### 2.1. miRNA Modulation of Signaling Pathways—Notch and TCL-1-Driven Activation of PI3K-AKT; NF-κB

AKT signaling has proved to be a key crossroad of several mutations and miRNAs in ALL. Activation of this pathway can enhance cell survival and, importantly, drug resistance (mainly to glucocorticosteroids and other drugs like **anthracyclins**, as shown in acute myeloid leukemia) [41,42]. Therefore, any modulation of AKT signaling can alter the response of lymphoblasts to these classes of chemotherapeutics. For example, both Notch signaling, which is deregulated in more than 60% of patients with T-ALL (similarly to ~50% of childhood T-ALL, which have activating mutations), and PTEN inactivation (found in more than 10% of patients) [43] can lead to AKT activation [44]. Certain translocations involving the T cell receptor (TCR) α/δ genes can also enhance the activity of this kinase. *TCL-1* (T cell leukaemia/lymphoma 1A oncogene) is frequently overexpressed due to chromosomal rearrangements of the TCR locus. TCL-1 can drive leukaemogenesis of T-ALL by interacting directly with AKT and enhancing its activity, which provides another means of activating this pathway [45]. **miR-19** is overexpressed in adult T-ALL and B-ALL samples compared to normal tonsillar lymphocytes. In one case, this was shown to be due to a translocation involving the TCR α/δ and the miR-17-92 locus (t(13;14)(q32;q11)). In a mouse model with transduced haemopoietic progenitor cells, Mavrakis et al. also demonstrated that Notch signaling in combination with **miR-19** ectopic expression could drive T-ALL. The authors also showed that the oncogenic role is believed to be due to the downregulation of *PTEN* and the pro-apoptotic *BIM* [33].

The set of miRNAs directly targeting ***PTEN*** in T-ALL cells has been expanded further to **miR-20**, **miR-26a**, **miR-92** and **miR-148/152** [34]. Of note, the crosstalk between different oncogenic pathways is highlighted by the finding that loss of this tumour suppressor gene can make Notch-driven leukaemia cells resistant to **gamma-secretase inhibitors** (GSIs) (which block Notch activation) [35], suggesting a potential, yet uninvestigated, role of AKT-modulating miRNAs in GSI drug resistance as well.

Interestingly, **miR-181a** is upregulated in primary T-ALL cells and correlates with AKT phosphorylation. Yan et al. also demonstrated that ectopic expression of this miRNA in HEK293 cells could activate the AKT signaling pathway and make cells less sensitive to **doxorubicin**. Furthermore, the upregulation of miR-181a was also shown upon treatment of Jurkat (childhood ALL) cells with doxorubicin, cyclophosphamide, cytarabine and cisplatin. Finally, knock-down of miR-181 with antagomirs in a dox-resistant subclone of Jurkat cells could re-sensitize the lymphoblasts to the chemotherapeutics mentioned above, showing the role of this miRNA in drug resistance [46].

The central role of **NF-κB** activation in apoptosis, cell survival and multidrug resistance has been demonstrated in several different types of cancer, including haematological malignancies like lymphomas, CLL and ALL [47,48,49,50]. For example, NF-κB can be activated in cells resistant to **vincristine** and **daunorubicin** and inhibit this transcription factor with BAY 11-7082, which blocks IκB degradation and resensitises murine T-ALL cells to these two chemotherapeutics [51]. The role of this family of transcription factors has been extensively discussed in ALL [52]. Therefore, miRNAs involved in the control of NF-κB may be implicated in drug resistance.

The human T cell leukaemia virus type 1 (HTLV-1) is the first described human retrovirus and can induce adult T cell leukaemia (ATL) in 1–5% of the individuals it affects [53]. The role of miRNAs in ATL has also been extensively studied [54]. One of the most interesting findings is that the virally encoded proteins TAX and HBZ (which normally transactivate proviral RNA expression and regulate 5′ LTR transcription) can deregulate some miRNAs. TAX’s ectopic expression has been shown to immortalise cells and activate NF-κB constitutively via direct interaction with IKKγ. It can also lead to the upregulation of **miR-21** (downregulates *PTEN*) and of the anti-apoptotic **miR-93** (suppresses *p21*) and **miR-155** (inhibits *TP53INP1*) [36]. Furthermore, **miR-31** is epigenetically silenced in primary ATL cells. The target of this miRNA is NIK, which promotes NF-κB signaling. Besides regulating this pathway, Yamigishi et al. found that miR-31 indirectly controls, through NF-κB, the levels of certain pro-apoptotic proteins BCL-XL, XIAP and FLIP [37].

### 2.2. miRNA Regulation of Apoptosis

While the activation of AKT and NF-κB by dysregulated miRNAs can tip the balance of cell processes in lymphoblasts from cell death to survival, some miRNAs can directly modulate apoptosis. Translocations of the immunoglobulin heavy chain (IGH) gene, leading to overexpression of various targets like MYC, BCL-2, etc., are commonly seen in B cell malignancies [38,55]. Interestingly, albeit rare in BCP-ALL, such a genetic perturbation involving IGH and **miR-125b** has been described [8,56]. This miRNA is known to downregulate **p53**, providing a possible underlying oncogenic effect of IGH translocation [38]. Of note, overexpression of miR-125b without IGH translocation has been described in adult ALL [8], and a similar translocation was detected in childhood ALL [57]. Furthermore, higher expression of this miRNA has been linked directly to B- and T-ALL leukemogenesis in mice [58], and to chemoresistance, relapse and poorer overall survival in paediatric ALL [59], therefore suggesting an important role in adult BCP-ALL as well.

Lastly, BCL-2 and MCL-1 are other antiapoptotic proteins that have been linked to chemotherapy resistance and relapse in a small cohort of ALL patients [60]. As mentioned above, miR-19 can be overexpressed in ALL cells, and it directly targets the mRNA levels of the pro-apoptotic BCL-2 member BIM [33]. This gene can also be downregulated by miR-20a, miR-27, miR-92 and miR-148/152, which are overexpressed in T-ALL cells and may contribute to poor drug response [34].

### 2.3. Other miRNA Targets—miRNAs in Glucocorticoid (GC) Resistance (GRα and MLL-AF4)

Perhaps the strongest link between miRNAs and drug resistance in ALL is found with **miR-142-3p**. This miRNA has been upregulated in a relatively large cohort of 74 T-ALL patients compared to normal peripheral blood cells (and in HTLV-1 T-ALL [61]). Lv et al. demonstrated that high expression of miR-142-3p is related to high-risk groups and poorer overall survival. The authors found that this miRNA can directly target the 3′ UTR of the glucocorticoid receptor (GRα). Upon antagomiR inhibition, cells (albeit the childhood T-ALL cell line CCRF/CEM) can be resensitised to dexamethasone [40].

Kotani et al. discovered that **miR-128b** and **miR-221** are frequently downregulated in MLL-translocation-driven ALL marked by poor prognosis and often by GC resistance. Interestingly, ectopic expression of these 2 miRNAs could resensitise cells to dexamethasone in the B-ALL cell line RS4;11 (and in the childhood B-ALL SEM cell line, too) and to etoposide, likely through the downregulation of the driver MLL-AF4 and the cell cycle regulator CDKN1b [39]. Therefore, AKT signaling, NF-κB activation, dysregulation of apoptosis and GR transcription may contribute to poor therapy outcomes in adult ALL (Figure 1).

## 3. Implication of miRNAs in Drug Resistance in Childhood ALL

ALL is the most common pediatric cancer, with an incidence of 20 cases per 1,000,000. It is manifested by increasing immature lymphoblast cells. There are two main subtypes of cALL depending on the lineage: B cell and T cell leukemia, with B cell prevalence around 75% of all cases. Other commonly used stratifications into risk groups are made according to clinical features and cytogenetic markers. The onset is typically at the age of 3 to 5 years. With the development of modern drugs and chemotherapy regimens, the treatment outcomes are substantially improved. Thus, the 5-year survival rates are significantly increased—from less than 60% in the 1980s to over 80% after 2004 and approaching 90% nowadays [12,62,63].

The standard treatment protocols include GCs, vincristine (vinca alkaloid), L-asparaginase, anthracyclines and antimetabolites (cytarabine, methotrexate, mercaptopurine) [64]. Despite the highly effective therapy and increased overall survival if relapse occurs, only 50% of patients achieve a second remission. This percentage is even lower if the relapse starts before the completion of primary treatment [65]. Thus, relapse and treatment failure appear to be the factors that dramatically affect the overall survival of childhood ALL patients. The intimate mechanisms behind these complications are not fully understood. The reasons are hidden in mutations in signaling pathways that affect apoptosis, proliferation, growth, and differentiation. In the last 15 years, the intense investigation of the role of miRNA in the pathogenesis and prognosis of childhood ALL allowed the identification of miRNA signatures related to prognosis, treatment outcome and resistance to some of the most commonly used drugs. In the light of the current review, we summarise the data on miRNA regulation of genes that ensure proliferation and cell growth through the activation of certain signaling pathways, and of genes that directly inhibit apoptosis of leukemic blast cells, and thus provide drug resistance.

### 3.1. miRNA Modulation of Signaling Pathways—AKT, Notch—NF-κB

Signaling pathways that control cellular processes such as proliferation, differentiation and apoptosis enhance cell survival of malignant cells. Different mutations may alter proteins that act as receptors, enzymes, or inhibitors. Signaling pathways are crosslinked at different levels, and their dysregulation may result in many unpredictable effects. miRNAs act as regulators of the translation of proteins involved in signaling pathways. Thus, alternations in miRNA expression are crucial factors that affect cellular processes. Molecules acting as inhibitors of different signal proteins are tested in vitro and in vivo to find specific drugs with less adverse effects and better treatment outcomes. In recent years miRNAs have been regarded as possible targets for developing new therapeutics [66,67].

In a cohort of 111 ALL patients, Li et al. found that the levels of **miR-99a** and **miR-100** are significantly downregulated compared to those in cells from healthy bone marrow donors. Among other targets (discussed below), these two miRNAs were demonstrated to suppress mTOR and IGF1R. The **AKT/mTOR** axis is frequently activated in childhood ALL and has been strongly implicated in GC resistance through balancing proliferation, autophagy, apoptosis and cell metabolism [68]. Furthermore, in paediatric T-ALL cell lines, AKT1 interacts directly with the GC receptor NR3C1 and phosphorylates it, making it inactive and ineffective GC therapy [69]. Therefore, not surprisingly, when these two AKT modulating miRNAs were overexpressed through mimics in 3 childhood ALL cell lines, lymphoblasts exhibited decreased proliferation and enhanced apoptosis in response to dexamethasone (likely due to the downregulation of the antiapoptotic BCL-2 family member *Mcl1*) [70].

Interestingly, another group (Moqadam et al.) described that **miR-99a** or/and **miR-100**, when co-expressed with miR-125b, could lead to **vincristine** resistance. It was shown in vitro that neither one of these miRNAs could cause insensitivity to this drug. Only the co-expression of miR-125b with miR-99a or miR-100 could trigger resistance to therapy. In the absence of vincristine in vitro, there was no cell cycle arrest nor increased apoptosis rate in miR-125b-, miR-99a-, or miR-100-expressing cells, suggesting a time- and context-specific role of these miRNA [71]. Further scientific data reported by Schotte et al. confirmed that these three miRNAs can be upregulated (14-25-fold) in childhood ALL patients with TEL-AML1 fusion (in 12 of 31 and 12 of 29 samples) and that their overexpression confers **vincristine** and **daunorubicin** resistance [72]. However, the exact mechanisms and targets of miR-99a, miR-100 and miR-125 remain uninvestigated.

Additionally, the same group reported that inhibited expression of **miR-454** correlated with L-asparaginase resistance [72]. This miRNA has been validated as a tumour suppressor in other malignancies like osteosarcoma, nasopharyngeal carcinoma, and multiple myeloma [73,74,75]. MiR-454 can induce apoptosis and suppress cell survival and proliferation by inhibiting c-Met, consequently inactivating the **AKT/mTOR pathway**. The c-Met/Akt/mTOR axis has been involved in drug resistance in vitro in other hematological malignancies [73].

Lin et al. demonstrated that the overexpression of miR-454 could restore sensitivity to cisplatin in nasopharyngeal carcinoma cells, confirming the role of this miRNA chemotherapy response [75]. On the opposite, it was shown by other groups that overexpression of miR-454 in the SGC-7901 gastric carcinoma cell line and HCT-116 colorectal cancer cells could mediate oxaliplatin resistance and promote proliferation through the CYLD gene. It regulates, in turn, the **NF-κB** and **TGF-β** signaling pathways and inhibits **PTEN** and activates the **AKT pathway** [76,77]. The same miRNA could exhibit opposite regulatory functions in certain cell types and tissues and use different pathways. Nevertheless, the exact involvement of miR-454 in L-asparaginase resistance in cALL is yet to be elucidated.

The same authors identified several dysregulated miRNAs in B-ALL samples, with **miR-708** overexpressed in high-risk patients. It was assumed that miR-708 upregulation might contribute to the initiation of relapse (which is partially in line with another study showing that levels of this miRNA increase in relapse compared to complete remission [78]). Notably, two of the targets of miR-708, *CNTFR* and *GNG12*, could explain at least some of the mechanisms that may lead to therapy failure. Downregulation of the first gene was shown to activate the **Jak/STAT** pathway, while the *GNG12* gene product is involved in **MAPK** signaling [79,80].

Another miRNA that affects drug sensitivity in T-ALL cells through modulation of a specific signaling pathway is the tumor suppressor **miR-101**. This miRNA was downregulated in T-ALL patients compared to control samples and directly targeted the 3‘UTR of Notch1. Ectopic expression in Jurkat cells through a mimic sequence decreased proliferation and enhanced apoptosis. Importantly, Qian et al. also proved that miR-101 could significantly attenuate drug resistance to doxorubicin in vitro [81].

Lastly, the NF-κB transcription factor family members are activated in childhood ALL [82]. NF-kB-driven transcription has been shown to induce resistance to doxorubicin and etoposide in CEM T-ALL cells [83]. Overexpression of **miR-125b** has been demonstrated to activate **NF-κB** by directly targeting the inhibitory protein TNF-α–induced protein 3 (TNFAIP3/A20) in Jurkat cells [84]. Interestingly, this miRNA can be translocated and overexpressed in BCP-ALL and correlates to chemotherapy resistance and poor survival in childhood ALL. This may result from the NF-kB activation as described above [59]. Another effect of miR-125b upregulation is the metabolic reprogramming of T-ALL in vitro. Liu et al. showed in Jurkat cells that high miRNA levels could lead to increased glucose consumption via upregulation of the glucose transporter GLUT1 and enhanced oxygen consumption [84]—events strongly associated with glucocorticoid resistance [85]. Thereby, there is considerable diversity in the pathways that miRNAs may dysregulate in their contribution towards drug insensitivity in childhood ALL (Table 2).

### 3.2. miRNA Regulation of Apoptosis

The dysregulation of apoptosis is crucial in developing different pathological conditions, including malignancies [92]. By regulating the expression of pro- and anti-apoptotic proteins, miRNAs directly affect cell survival and drug sensitivity. Alternations in apoptosis are detected in childhood ALL as well. Higher expression of the anti-apoptotic protein BCL-2 and downregulation of Bax were related to relapse and bad prognosis [93].

A direct link between miRNAs and apoptosis upon drug treatment in childhood ALL is provided by **miR-34a** (also discussed below in CLL). Najjary et al. demonstrated in Jurkat cells that expression of this miRNA alone or in combination with doxorubicin treatment could decrease *BCL-2* and increase *Caspase-3* and *p53* levels, thus significantly enhancing cell death [88]. Another piece of evidence of the importance of miRNAs in apoptosis is provided by Jiang et al., who reported that **miR-652-3p** was significantly downregulated in pre-B pediatric ALL patients at diagnosis and relapse but upregulated at remission and in healthy control samples. Atopic expression of an antagomiR sequence in REH and RS4;11 leukemic cells induced apoptosis and made cells more sensitive to vincristine and cytarabine, but the target genes have been identified neither in childhood ALL patients nor in cell lines [87]. Lastly, **miR-204** is silenced by promotor methylation in ALL. This miRNA regulates *Irak1,* which mediates NF-κB activation and consequent transcription. Importantly, these miR-204 targets also regulate the stability of the antiapoptotic protein MCL1, which has been linked to response to vincristine [86]. Therefore, not surprisingly, ectopic expression of a miR-204 mimic in Jurkat cells could lead to decreased proliferation and apoptosis [94]. All these proofs of the miRNA-mediated control of apoptotic proteins show that the dysregulation of programmed cell death could provide survival advantages and mediate drug resistance in childhood lymphoblastic leukemia.

### 3.3. Other miRNA Targets—MDR1, Proteasomal Degradation, GC Resistance (MLL-AF4 and GR Modulation)

A study comparing matched samples at diagnosis and in complete remission (CR) found that two miRNAs are highly expressed when therapy is successful—**miR-223**, which targets *E2F1* (involved in DNA damage response, cell cycle arrest and apoptosis), and **miR-27a**, described to regulate the epigenetic modifier BMI1 [78]. Consistently, another group demonstrated that downregulation of the latter miRNA and miR-331–5p correlates with a higher risk of relapse in a mixed cohort of ALL and AML patients. It was shown in vitro (in HEK-293 and AML cell lines) that both miRNAs control the levels of the oncoprotein **MDR1** (multiple drug resistance protein 1). Importantly, ectopic expression of miR-27a or miR-331-5p could restore sensitivity to doxorubicin in vitro [89].

Interestingly, Kumar et al. identified miR-223 as a **Notch** and **NF-κB** signaling pathway in T-ALL. This miRNA appeared to downregulate the *FBXW7* tumor suppressor gene involved in ubiquitination and proteasomal degradation of oncoproteins. The authors showed that miR-223 has implicated in GSI (γ-secretase inhibitor) resistance and that antagomiRs could induce sensitivity in GSI-resistant T-ALL cells in vitro [90].

As mentioned previously, for adult ALL, **miR-128b** and **miR-221** are strongly implicated in glucocorticoid resistance. They are found to be downregulated in MLL-rearranged ALL compared to other childhood ALL subtypes. These miRNAs directly target the mRNAs of the fusion gene and thereby downregulate *CDKN1B*, which generates the p27- cell-cycle checkpoint regulator. Importantly, the restored expression of both miRNAs sensitised the childhood ALL cell lines SEM to dexamethasone [39].

Li et al. reported that in high-risk childhood ALL patients (WBC > 5 × 10^4^, T-ALL, MLL-rearranged gene, BCR-ABL fusion gene), **miR-99a** and **miR-100** are downregulated. Further, they described a novel target inhibited by both miRNAs—*FKBP51*. As a result of the reduced expression of miR-99 and miR-100, FKBP51 levels are high and suppress the activation of the glucocorticoid receptor (GR). Therefore, ectopic expression of these miRNAs would restore GR function upon dexamethasone treatment and lead to reduced proliferation and augmented apoptosis [70].

Lastly, even if the role of **miR-708** in childhood ALL may seem to be controversial or perhaps cell-type or stage-specific, Han et al. demonstrated that high levels of this miRNA (as well as miR-223 and miR-27a) correlate with higher relapse-free survival rates and good response to 7-day prednisolone monotherapy. Notably, **miR-708** can downregulate *FOXO3* levels both in vivo and in vitro. This transcription factor is a direct target of the GR [91] and is further known to be involved in GC target gene expression [95]. Therefore, not surprisingly, a positive correlation of this miRNA with GC sensitivity was observed—miR-708 was highly expressed in childhood ALL patients with a good response to 7-day prednisolone monotherapy. It was downregulated in the prednisolone poor response group [78].

In summary, miRNAs have been shown to control many signaling pathways connected to proliferation and survival, several transcription factors, and specific key apoptosis mediators, thus leading to drug resistance (Figure 2).

## 4. Implication of miRNAs in Drug Resistance in CLL

CLL is the most common type of leukaemia, accounting for ~1% of all cancers and about 10% of all haematological malignancies [12]. B cell CLL (>95% of cases) is characterised by clonal expansion and accumulation of immature CD5+/CD19+ B cells in the bone marrow and lymph nodes, where oncogenesis is believed to be driven by microenvironment signals, and in the peripheral blood [96]. The most clinically significant markers for poor disease outcomes are a lack of mutations in the immunoglobulin heavy chain variable region locus (IgHV), deletion of chromosome 13q (affecting the pro-apoptotic protein BCL-2) or 17p (affecting the tumour suppressor p53), as well as high expression of ZAP-70 and CD38 [13]. Standard chemotherapies, including fludarabine, cyclophosphamide and the targeted anti-CD20 antibody rituximab, achieve 10-year overall survival of over 70%. However, the prognosis for relapse and high-risk patients can be poor—less than 25% 5-year survival [97]. Differential expression between malignant and normal B cells and between CLL samples with different cytogenetics have linked hundreds of miRNAs to the pathogenesis, progression and prognosis of CLL [19,23]. Most of them implicated in drug response are shown in Table 3.

### 4.1. miRNA Modulation of Signaling Pathways—BCR, NF-κB, TLR and TCL-1-AKT

**B cell receptor (BCR) signaling** has been validated as an essential factor in the pathogenesis of CLL (and other B cell malignancies like diffuse large B cell lymphoma), providing stimulation of critical pro-survival pathways like PI3K-AKT and NF-κB [116]. Understanding the importance of such pathways has led to the development of small-molecule inhibitors targeting signal transducers such as BCR-associated kinase (BTK), SYK and PI3Kδ. For example, ibrutinib (a BTK inhibitor) shows remarkable response rates of nearly 80% [116]. Significantly, the expression of certain mediators of BCR signaling in CLL can be regulated by miRNAs. Thus, by investigating over 700 miRNAs in 168 patient samples, **miR-150** was found to inversely correlate with unmutated IgHV and ZAP70 levels (BCR heavy chain and enhancer of BCR signaling, respectively). The frequently deleted cluster of **miR-34b/c** located on chromosome 11q can also target ZAP70 [106]. Furthemore, miR-150 has been shown to directly regulate the levels of two BCR signaling mediators—the transcription factor FOXP1 and the PI3K adaptor molecule GAB1. Of note, lower levels of miR-150 and higher expression of the abovementioned proteins correlate with poorer overall survival [108]. Similarly, **miR-155** targets SHIP1, which can inhibit BCR signaling. The high expression of this miRNA was found to correlate with poor treatment-free and OS in a cohort of over 260 CLL patients. This finding again suggests the importance of BCR signaling in cell survival and therapy response [109]. BCR signaling and the downstream activation of PI3K and NF-κB pathways have not only been implicated in proliferation and survival, but also in the response to chemotherapy. For instance, the activation of AKT1 in particular has been shown to lead to fludarabine resistance in CLL cells (through STAT3 signaling) [117].

**NF-κB** is overactivated in CLL, leading to enhanced survival and decreased apoptosis due to fludarabine and dexamethasone treatment [118,119]. Rearrangements resulting in BCR-ABL fusion and involving BCL-3, which stimulates the p50 and p52 subunits of NF-κB, can also activate this transcription factor [120,121]. Strong DNA binding of the RelA subunit of NF-κB has been shown to correlate to tumour burden (cell doubling time and cell counts) in patients, in vitro cell survival and resistance to fludarabine [122]. Several miRNAs are known to regulate NF-κB activation in hematologic malignancies [99]. Of note, **miR-15a** and **miR-16-1**, deleted in more than 60% of CLL, have been shown to suppress IKKα. It phosphorylates the inhibitor of NF-κB (IkB), targeting it for ubiquitination and proteasomal degradation, which activates NF-κB signaling [99]. Members of the **miR-9** family (miR-9-3) can be silenced in CLL patients and cell lines by methylation and are implicated in NF-κB activation [98], likely through direct targeting of the 3′ UTR of the p50 subunit of NF-κB [99]. Similarly, **miR-708** has been found to downregulate IKKβ, which suppresses NF-κB signaling directly. However, this miRNA can be epigenetically silenced in CLL cells. Baer et al. proved that methylation of the **miR-708** enhancer correlates with poor treatment-free survival in a cohort of nearly 300 patients. The findings are similar to what has been detected in childhood ALL and response to glucocorticoids [78], suggesting the role of this miRNA and NF-κB signaling in drug response and relapse [123]. Furthermore, NF-κB activation can be regulated by a negative feedback loop involving **miR-146a** [107]. This miRNA’s downregulation (or knockout) can lead to earlier onset and more aggressive B cell malignancies in c-Myc-driven in vivo models [124]. Its levels are deregulated in CLL patients, as well [125].

**Toll-like receptors (TLRs)** have been established as essential co-activator molecules in B cells. The importance of TLR signaling has been investigated in CLL, revealing that stimulation of this receptor family can protect from apoptosis and correlates (through activation of NF-κB and STAT3) with poor prognosis [126,127]. Interestingly, TLR signaling in the microenvironment of CLL cells could activate **NF-κB** and has been implicated in CLL cell proliferation and survival [127] by activating members of the **miR-17~92** family [128]. Importantly, it has been shown that TLR signaling can rescue cells from apoptosis upon fludarabine treatment. **miR-155-3p** levels have risen concomitantly, implicating this miRNA in drug resistance [129].

The **protooncogene *TCL-1*** (T cell leukemia/lymphoma 1) is frequently overexpressed in T-ALL [45] and CLL with chromosome 11q deletions [130]. Furthermore, transgenic mice expressing *TCL-1* in B cells develop CLL, which demonstrates the leukemogenic properties of this gene [131]. Interestingly, this mouse model has served to study drug efficacy. Johnson et al. proved that TCL-1 is connected with fludarabine resistance in a p53-independent manner [132]. Furthermore, TCL-1 has been shown to interact directly and activate **AKT** in both T-ALL and CLL cells [133,134]. In particular, Hofbauer et al. have also demonstrated that targeting TCL-1 by siRNA can reduce AKT activation and resensitise resistant cells to fludarabine [134]. Therefore, it is noteworthy that **miR-29** and **mir-181** can directly regulate *TCL-1* expression in CLL patients, implicating them (at least indirectly) in drug resistance [104]. Of note, miR-29 has been confirmed to directly target AKT (among a few other targets) [105]. **miR-22** can also activate AKT signaling in CLL and enhance cell proliferation [101]. However, the role of this miRNA in fludarabine resistance has not been studied yet.

Comparing expression in responders and non-responders to this drug, high levels of three other miRNAs have been strongly implicated in refractory disease—**miR-21**, **miR-148a** and **miR-222**. Significantly, anti-miRNA oligonucleotides against miR-21 and miR-222 enhanced fludarabine-driven apoptosis suggesting a therapeutic window [100]. The mechanism of how these miRNAs make CLL cells refractory to treatment has not been elucidated yet, but Ferracin et al. suggest that it may be through regulation of PTEN and/or the cell cycle [100]. The high expression of miR-21 in non-responders has been confirmed by another study, which also highlighted that increased levels of **miR-34** predict an excellent response to fludarabine [135]—proof in line with previous studies presenting this miRNA as downregulated in fludarabine-refractory cells [136,137].

Lastly, PI3K-AKT signaling can be controlled by the tumour suppressor PTEN, downregulated in CLL and may also have prognostic value [138]. Zou et al. demonstrated that **miR-26a**, which has previously appeared overexpressed in CLL [102], and **miR-214** could directly downregulate *PTEN* mRNA levels. High expression of miR-26a correlates to inferior time to first treatment [103], which provides another 2 miRNAs that may be implicated in PI3K-AKT-mediated leukemogenesis.

### 4.2. miRNA Regulation of Apoptosis and Autophagy

Besides the modulation of signaling pathways that augment cell proliferation and survival, miRNAs can directly alter the levels of proteins involved in processes like apoptosis and autophagy, serving as rescue mechanisms for CLL cells upon chemotherapy treatment. Interestingly, miRNAs are frequently targeted by deletions or translocations, leading to lower expression, suggesting a tumour suppressor role of these miRNAs. This has been shown for **miR-15a** and **miR-16-1**, which target the antiapoptotic protein **BCL-2**, in over 60% of CLL patients [9]. Cimmino et al. found that both miRNAs are indeed inversely correlated to Bcl-2 gene expression in a cohort of 26 CLL patients. Therefore, downregulation of these miRNAs protects from apoptosis, while overexpression of miR-15a or miR-16-1 would directly diminish BCL-2 levels and induce apoptosis [110]. Such antisense strategies targeting BCL-2 have proven therapeutic potential in several malignancies, including re-sensitizing cells to chemotherapy [139] (discussed later). Similarly, lower expression of **miR-29**, described in *p53* mutant CLL samples [111], could result in higher levels of the anti-apoptotic protein MCL-1 and again in evasion of cell death [112].

A microarray covering over 800 miRNAs detected the downregulation of a number of these regulatory molecules compared to normal peripheral blood B cells in a cohort of 156 CLL patients. Interestingly, mimics of some of these hits were able to induce apoptosis in primary CLL cells. Zhu et al. also demonstrated that **miR-181a** and **b** directly downregulate the mRNA levels of the antiapoptotic proteins BCL-2, MCL-1 and XIAP. Lastly, the authors showed that forced expression of miR-181 (as well as **miR-15**, **16** and **34**) could enhance **fludarabine**-induced apoptosis in primary CLL cells (from 40 samples) likely in a p53-dependent manner [102]. In contrast (and in another context), when looking at a smaller cohort of 39 patients after their first 6 fludarabine cycles, Moussay et al. found that miR-181a is upregulated in refractory cells (making the role of this miRNA in drug response in CLL controversial). In addition, **miR-221** was also upregulated in resistant patients, while **miR-29** was downregulated (in accordance with previous studies) [114].

**TP53 (p53)** is a master regulator of DNA damage response, apoptosis, the cell cycle and gene expression. Deletions and/or mutations of this gene are well-established to correlate to worse overall survival and poor response to chemotherapeutics (like chlorambucil, Cytoxan, prednisolone, vincristine and fludarabine) in CLL [140,141]. Interestingly, the roles of miRNAs and p53 in apoptosis and drug response may be reversed—p53 can control the expression of miRNAs, most likely through transactivation of their expression by binding to predicted upstream DNA regulatory elements., This, in turn, would determine cell fate upon chemotherapy. The first study looking at the interplay between p53 and miRNAs demonstrated that **miR-17-5p**, **miR-29** and **miR-34a** are downregulated in patients with p53 mutations, who have a poorer prognosis. These three regulatory transcripts have been connected with the control of apoptosis (through BCL-2 and MCL-1) and the cell cycle (through CKD4, CDK6, E2F3, cyclin E and c-Myc) [111]. The already mentioned miR15a/16-1 cluster is regulated by p53. These findings demonstrate that deletion and/or mutation of the “guardian of the genome” affect a number of miRNAs, which in turn regulate apoptosis, BCR and NF-κB pathways—all linked to drug resistance as described above [106], in particular to fludarabine [136,137].

**Autophagy** has been well-established as a survival mechanism in malignant cells under different forms of stress, such as starvation and chemotherapy. Interestingly, CLL cells treated with fludarabine or a PI3Kδ inhibitor (CAL-101) undergo autophagy [142]. Indeed, inhibition of specific mediators of this process (autophagosome-lysosome fusion and the AMPK/ULK1 pathway) could enhance the cell-killing effect of inhibitors such as of CDKs (flavopiridol) [142] or BCL-2 (venetoclax) [143]. **miR-130a** is epigenetically silenced in CLL. Kovaleva et al. have confirmed its role in autophagy regulation [144,145]. Overexpression of this miRNA could downregulate the autophagy-related gene *ATG2B*, which enhanced apoptosis following starvation. Therefore, even if the role of this miRNA in drug resistance has not been exploited in CLL, its deregulation may provide lymphocytic cells with a survival mechanism upon chemotherapy treatment. This hypothesis could be supported by evidence from ovarian cancer where miR-130a is related to paclitaxel and cisplatin resistance [113]. Thereby, AKT and BCR signaling, NF-κB activation, modulation of apoptosis and the expression of CD20 (see below) constitute the main targets of miRNAs, by which CLL cells achieve drug resistance (Figure 3).

### 4.3. Other miRNA Targets—CD20 and Resistance to Rituximab

Interestingly, **miR-125b** and **miR-532-3p** have been described to correlate to lymphocytopenia following rituximab treatment inversely. A bioinformatics approach with putative targets demonstrated that these miRNAs might lead to therapy resistance through downregulation of the direct target of this antibody-based treatment—CD20 (*MS4A* genes) [115].

## 5. Perspectives of miRroring Strategies in Targeting of Drug Resistance in Lymphocytic Leukaemias

The importance of miRNAs for the pathogenesis and prognosis of lymphocytic leukaemias has been established previously, but their involvement in drug response has not been discussed thoroughly. Here we summarise the scientific evidence which demonstrates the role of these post-transcriptional mRNA modulators in drug resistance. Activation of AKT or NF-κB (Figure 4) can provide pro-survival signals. Direct control of apoptosis appears as the major escape mechanism employed by dysregulated miRNAs in relation to chemotherapy response in ALL and CLL. These data are not unexpected given the germane part these pathways play in relapses and poor survival (described above). Interestingly, we also found that certain miRNAs are implicated in several other dysfunctional molecular axes contributing to drug resistance, such as glucocorticoid uptake and receptor activation, transduction through TLRs, transcription by MLL-AF4, and regulation of CD20 levels.

### 5.1. Silencing of miRNAs

There is a great amount of ongoing work (including in clinical trials) related to the implementation of various approaches aiming to silence certain miRNAs (when they are oncogenic—“oncomiRs”) or to express them ectopically (when they act as tumour suppressors). Such therapeutic strategies have been under development for ~20 years. They include inhibiting oncomiRs through complementary binding of various chemically-modified oligonucleotides like **antagomiRs** and anti-miR oligonucleotides (**AMOs**) [146,147]. Furthermore, microRNA **sponges**, which can simultaneously block a family of miRNAs (or several different ones), can also be engineered and have shown therapeutic potential in preclinical studies [148]. Small-molecule inhibitors of miRNAs (SMIRs) impeding different steps in the biosynthesis or target-binding of miRNAs have also proven a valid strategy [147]. Lastly, there are also “double-edged sword” approaches—either using a combination of small interfering RNAs (siRNAs) and miRNAs aiming at a single target [149] or employing several miRNAs covering different players in the same pathway (e.g., PI3K-RAS-RAF) [150].

In ALL and CLL, we have highlighted that high expression of miRNAs such as miR-17~92, miR-155-3p, miR-21, miR-221, miR-222, and others is strongly associated with drug resistance to, e.g., fludarabine, making them potential therapeutic targets for inactivation. Several studies provide encouraging examples of the beneficial effect of miRNA inhibition on chemotherapy treatment in vitro. Yan et al. demonstrated that **miR-181a** is overexpressed in T-ALL samples, and its silencing by antagomiRs can resensitise Jurkat cells to doxorubicin cyclophosphamide, cytarabine and cisplatin [46]. Harada et al. used a locked-nucleic acid (LNA) antisense approach targeting **miR-17** in cALL cells. Silencing this miRNA could enhance the response to dexamethasone by increasing BIM levels and inducing apoptosis [151]. Similarly, LNA anti-**miR-21** and **-222** augment cell death induction by fludarabine in a CLL model cell line [100]. Notably, the therapeutic power of antagomiRs has been demonstrated in vivo as well. In an animal model using MEC-1 CLL cells, injecting an **anti-miR-17** oligonucleotide leads to reduced tumour burden and increased survival in mice [152].

### 5.2. Ectopic Expression of miRNAs

Another therapeutic approach is to re-express tumour suppressor miRNAs. This strategy is readily achievable in preclinical models through the intracellular introduction of exogenous synthetic double-stranded RNA **mimics** (via lipid or polymeric-based carriers), viral vectors expressing the desired oligonucleotide, and other approaches [146,147,153]. Here, we pinpoint the involvement in the therapy failure of several miRNAs. Thus, **miR-15a/16-1** and **miR-34b/c** can be absent in CLL due to deletions in chromosomes 13q and 11q, respectively, while **miR-29**, **miR-181a** and **b**, and others have reduced expression in CLL. Again, several in vitro studies can serve as proof-of-concept research for the significant therapeutic advantage of miRNA modulation combined with standard chemotherapy. Ectopic expression of **miR-128b** and **miR-221** in RS4;11 and SEM ALL cells can be downregulated in MLL-AF4-driven ALL. This event resensitises the lymphoblasts to etoposide and dexamethasone [39]. Similarly, **miR-101** is downregulated in T-ALL samples compared to control cells, and its levels decrease upon doxorubicin treatment in vitro. Interestingly, Qian et al. demonstrated that this miRNA directly targets Notch1, and overexpressing a mimic in Jurkat cells enhance chemotherapy-induced sensitivity and apoptosis [81]. As mentioned above, **miR-181** is downregulated in CLL, most likely because it can decrease the levels of several anti-apoptotic BCL-2 members. Ectopic expression of a mimic in primary cells can enhance fludarabine-driven apoptosis [102].

Lastly, a number of miRNAs are epigenetically silenced in lymphocytic leukaemias, including **miR-130a**, **miR-708**, **miR-143** and **miR-31**. Therefore, it is possible to re-express such miRNAs through epigenetic modifiers. A successful example is the use of HDAC inhibitors to activate the transcription of *DICER*, which HTLV-1 suppresses in ATL. This could lead to restored processing of several miRNAs and enhanced sensitivity to doxorubicin and etoposide, even if the approach is non-specific [154]. Therefore, several therapeutic strategies could target miRNAs and drug resistance in lymphocytic leukaemias (Figure 5).

Dysregulated pathways in ALL and CLL (left-hand side) and a list of therapeutic strategies targeting miRNAs that can resensitise malignant lymphocytes to chemotherapy and induce apoptosis (right-hand side).

### 5.3. Promises and Drawbacks

Undoubtedly, miRNAs cover most of the requirements for ideal biomarkers [155]. Although the studies showing their therapeutic potential have increased in the last few years, the number of miRNAs included in trials that eventually reached the clinical bench remains limited. Mostly, RNA-based drugs are designed to bind to a specific target or group of targets. However, miRNAs can regulate numerous tissue or cell-specific transcripts that belong to the same or different biological pathways [156,157]. Conversely, the limited length of eight nucleotides through which miRNA drugs interact with the seed sequence is not strongly unique for the specific targets. Consequently, unspecific binding and regulation of undiscovered transcripts are quite possible, leading to off-target effects with unanticipated and adverse therapeutic consequences [158,159]. In addition, miRNA:mRNA associations are incompletely understood, and high seed sequence similarity does not necessarily guarantee physiological regulation of the mRNA target [160,161]. Functional experiments in both in vitro and in vivo models are required to assess miRNAs’ suitability as therapeutic molecules. The number of bioinformatically predicted miRNA:mRNA interactions is currently poorly validated in experimental systems [162,163]. Even though many high-throughput techniques facilitate in vitro validation, studies in animal models have significant throwbacks, such as the impossibility of completely mimicking the pathology of interest and interference with endogenous miRNAs. Another obstacle is that some miRNAs, such as Let-7, could modulate the immune response in vivo by interacting with transcription factors and other regulators of intercellular communication pathways [164]. One of the reasons for chemotherapy failure is the resistance that some tumors develop against particular drugs. The primary treatment with let-7g-containing lentivirus suppressed tumor number and area in a murine model bearing non-small cell lung cancer xenografts [165]. However, the prolonged treatment appeared insufficient, as shown by the development of tumor relapse [165] due to the loss of the let-7-binding site in the 3′UTRs of some oncogenes such as HMGA2 [166].

Another drawback of miRNA-based therapies is illustrated by problems connected to the fact that RNA drugs are modified by the selective incorporation of 2′-O-methyl (2′OMe) in uridine or guanosine nucleosides to avoid the degradation caused by nucleases, as well as to improve efficiency and target specificity [167,168]. It has been found that even though such a modification avoids cytokine production and off-target effects, it causes toxicity or renders the molecule less efficient [159].

One requirement that stays at the front of miRNA replacement therapy is effective and safe drug delivery. The excessive application of the drug might cause hepatotoxicity, followed by organ failure and death [169]. This ultimately limits the amount of the required drug as it is frequently above the levels a cell can tolerate [169,170,171,172,173,174]. Moreover, the blood-brain barrier significantly restricts RNA-based drug delivery to tumors in the central nervous system. This scenario can be overcome with physical activities such as intrathecal delivery [175]. These examples illustrate that the application of miRNA-based therapy in clinical practice is a challenge demanding further efforts and improvements until its routine use is launched.

In summary, miRNAs are strongly implicated in drug resistance in lymphocytic malignancies. Increasing evidence demonstrates that targeting these small regulatory oligonucleotides in cell lines or mouse models can induce cell death and enhance drug response. With the advancement and required refinement of novel therapeutic strategies and delivery systems like those in mRNA-based vaccines, more studies will likely be investigating the potential benefits of systemic administration of antagomiRs or miRNA mimics as single agents and in combination with immune-chemotherapeutics. Therefore, despite all the challenges mentioned above, looking at the miRror of drug resistance in adult and childhood ALL and CLL reveals an upcoming period of a promising multidisciplinary effort to overcome refractory disease and relapses and improve patient survival.

## Figures and Tables

**Figure 1 ijms-23-04657-f001:**
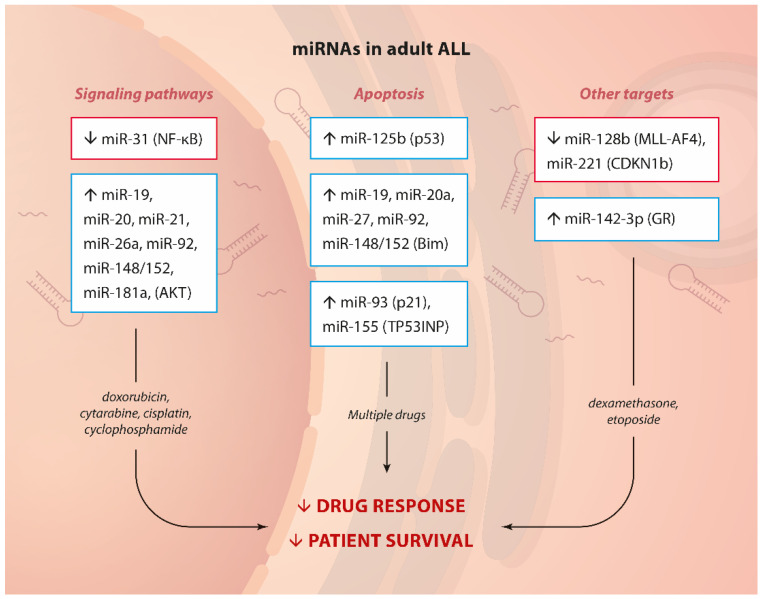
miRNA and drug resistance in adult ALL.Summary of up- or down-regulated miRNAs, their targets or the pathways affected (in brackets) and involvement in drug resistance in adult ALL.

**Figure 2 ijms-23-04657-f002:**
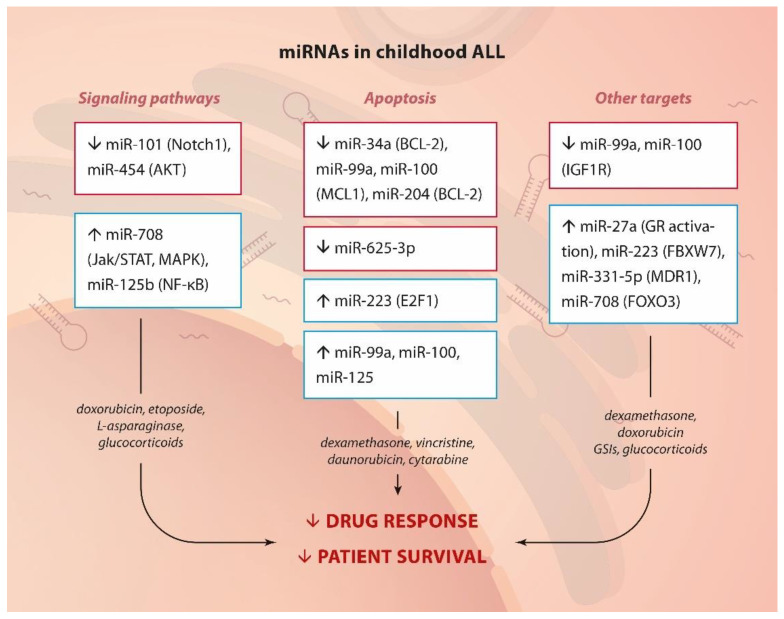
miRNA and drug resistance in childhood ALL.Summary of up- or down-regulated miRNAs, their targets or pathways they affect (in brackets) and their involvement in drug resistance in childhood ALL.

**Figure 3 ijms-23-04657-f003:**
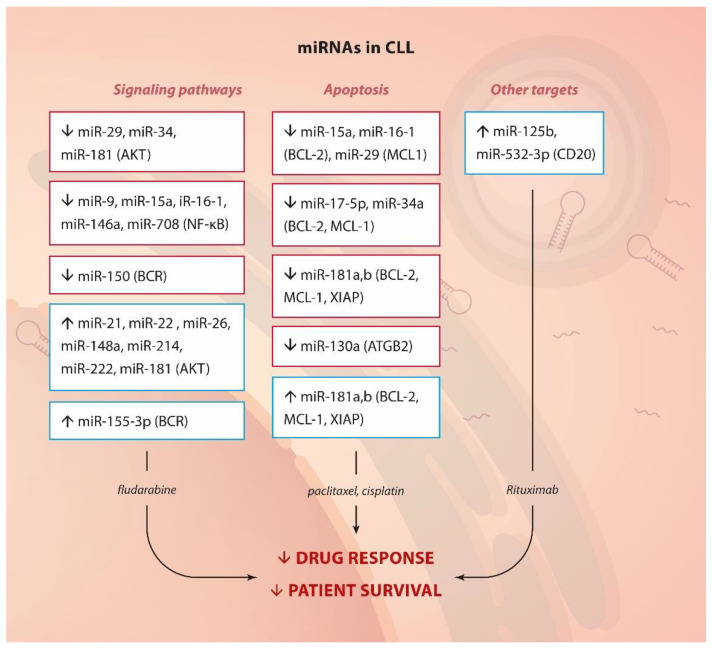
miRNA and drug resistance in CLL. Summary of up- or down-regulated miRNAs, their targets or pathways they affect (in brackets) and their involvement in drug resistance.

**Figure 4 ijms-23-04657-f004:**
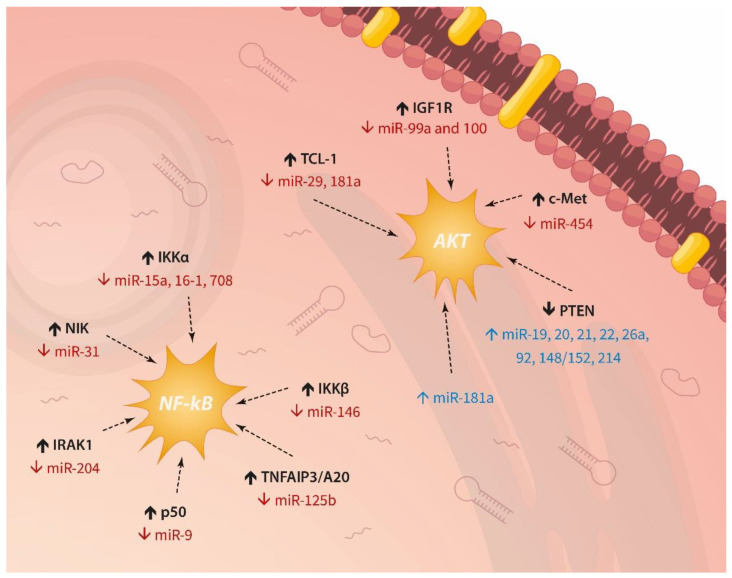
Recurrent miRNA-driven activation of AKT signaling and NF-κB in lymphocytic leukaemias.

**Figure 5 ijms-23-04657-f005:**
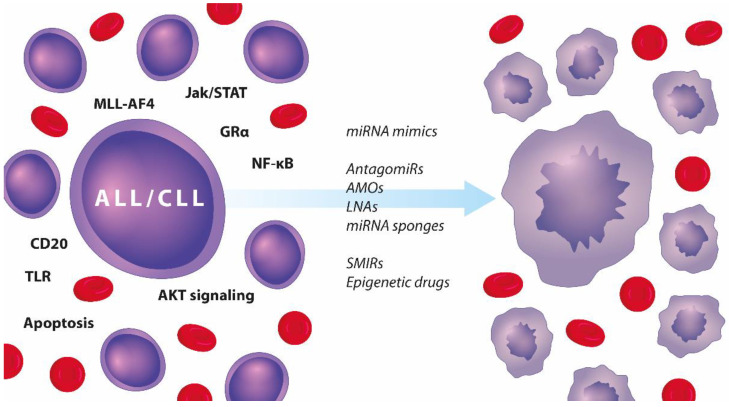
Summary of therapeutic approaches to targeting the miRror in adult and paediatric ALL and CLL.

**Table 1 ijms-23-04657-t001:** miRNAs, their targets and their effects in adult ALL.

miRNA	Targets/Pathways	Outcome/Correlation	Reference
** *Signaling pathways* **			
↑miR-19	PTEN (AKT)	Promotes oncogenesis	[33]
↑miR-20a	PTEN (AKT)	Promotes oncogenesis; potential role in resistance to GSIs	[34,35]
↑miR-21	PTEN (AKT)	Potential role in resistance to GSIs	[35,36]
↑miR-26a	PTEN (AKT)	Promotes oncogenesis; potential role in resistance to GSIs	[34,35]
↓miR-31	NIK (NF-κB)	Increased cell survival; reduced apoptosis	[35,37]
↑miR-92	PTEN (AKT)	Promotes oncogenesis; potential role in resistance to GSIs	[34,35]
↑miR-148/152	PTEN (AKT)	Promotes oncogenesis; potential role in resistance to GSIs	[34,35]
** *Apoptosis* **			
↑miR-19	BIM	Promotes oncogenesis	[33]
↑miR-20a	BIM	Promotes oncogenesis; potential role in resistance to GSIs	[34]
↑miR-27	BIM	Promotes oncogenesis; potential role in resistance to GSIs	[34]
↑miR-31	BCL-XL, XIAP, FLIP (through NF-κB)	Resistance to apoptosis	[37]
↑miR-92	BIM	Promotes oncogenesis; potential in resistance to GSIs	[34]
↑miR-93	p21	Uninvestigated	[36]
↑miR-125b	p53	Potential role in leukemogenesis	[38]
↑miR-148/152	BIM	Promotes oncogenesis; potential role in resistance to GSIs	[34]
↑miR-155	TP53INP1	Uninvestigated	[36]
** *Other targets* **			
↓miR-128b	MLL-AF4, CDKN1b	Resistance to GCs and etoposide	[39]
↑miR-142-3p	GRα	Resistance to GC/high-risk groups poorer overall survival	[40]
↓miR-221	MLL-AF4, CDKN1b	Resistance to GCs and etoposide	[39]

Arrows (↓ and ↑) indicate miRNA downregulation or upregulation.

**Table 2 ijms-23-04657-t002:** miRNA, their targets and their effects in childhood ALL.

miRNA	Targets/Pathways	Outcome/Correlation	Reference
** *Signaling pathways* **			
↓miR-101	Notch1	Resistance to doxorubicin	[81]
↑miR-125b	TNFAIP3/A20 (NF-κB)	Poor survival (doxorubicin, etoposide, glucocorticoids? in vitro)	[59,84,85]
↓miR-204	IRAK1 (NF-κB)	Resistance to vincristine	[86]
↓miR-454	c-MET (AKT)	Resistance to L-Asparaginase (cisplatin in nasopharyngeal sarcoma)	[72,75]
↑miR-708	CNTFR (Jak/STAT), GNG12 (MAPK)	High-risk group and relapse (multiple drug resistance)	[78]
** *Apoptosis* **			
↓miR-99a and -100	IGF1R, mTOR (MCL1)	High risk; dexamethasone resistance	[70]
↑miR-99a, miR-100 and miR-125 (together)	Uninvestigated	Resistance to vincristine and daunorubicin	[71,72]
↑ miR-223	E2F1	Complete remission (in B-ALL and AML)	[78]
↓miR-652-3p	Uninvestigated	Relapse; resistance to vincristine and cytarabine	[87]
↓miR-34a	BCL-2	Resistance to doxorubicin	[88]
↓miR-204	MCL1 (through IRAK1 and NF-κB)	Resistance to vincristine	[86]
** *Other targets* **			
↑miR-27a	BMI1 (epigenetic modifier) MDR1	Complete remission Sensitivity to doxorubicin (in K562 cell lines) Low expression in relapse patients	[78,89]
↓miR-99a and -100	FΚBP51 (and GR expression and activity)	Downregulated in high risk; Dexamethasone resistance	
↑miR-223	FBXW7 (ubiquitination)	Resistance to GSIs in T-ALL	[90]
↑miR-331-5p	MDR1	Sensitivity to doxorubicin (in K562 cell lines) Low expression in relapse patients	[89]
↑miR-708	FOXO3 (self-renewal, AKT activation in myeloid cells)	Good response to GCs; relapse-free survival	[91]
↓miR-128b and miR-221	MLL-rearranged fusion gene (thereby downregulating *CDKN1B*, and p27)	Resistance to GCs	[39]

Arrows (↓ and ↑) indicate miRNA downregulation or upregulation.

**Table 3 ijms-23-04657-t003:** miRNAs, their targets and their effects in CLL.

miRNA	Targets/Pathways	Outcome/Correlation	Reference
** *Signaling pathways* **			
↓miR-9	NF-κB	Advanced Rai stage	[98]
↓miR-15a and miR-16-1	IKKα (NF-κB)	Resistance to fludarabine and dexamethasone	[99]
↑miR-21	PTEN (AKT)	Resistance to fludarabine	[100]
↑miR-22	PTEN (AKT)	B-CLL cell proliferation	[101]
↑miR-26	PTEN (AKT)	Advanced Binet stage; inferior time to first treatment; resistance to apoptosis in vitro	[102,103]
↓miR-29	TCL-1, AKT (AKT)	Indirect implication in resistance to fludarabine (through TCL-1)	[104,105]
↓miR-34b/c	ZAP70 (B cell receptor signaling)	Poor overall survival	[106]
↓miR-146a	NF-κB	Earlier onset of B cell malignancies (in mice)	[107]
↓miR-150	ZAP70, FOX1, PI3K/GAB1 (B cell receptor signaling)	Poor overall survival	[106,108]
↑miR-155	SHIP1 (B cell receptor signaling)	Poor therapy outcome and overall survival	[109]
↓miR-181a, b	TCL-1 (AKT)	Indirect implication in resistance to fludarabine (through TCL-1)	[104]
↑miR-214 and	PTEN (AKT)	Resistance to apoptosis in vitro	[103]
↓miR-708	IKKα (NF-κB)	Poor treatment-free survival	[78]
** *Apoptosis* **			
↓ miR-15a and miR-16-1	BCL-2	Overexpression induces apoptosis in vitro	[102,110]
↓miR-17-5p	BCL-2	Poor prognosis (correlates with p53 inactivation)	[111]
↓miR-29	MCL1	Poor prognosis (correlates with p53 inactivation)	[111,112]
↓miR-34a	BCL-2, CDK4, CDK 6, E2F3, Cyclin E, c-Myc	Poor prognosis (correlates with p53 inactivation)	[102,111]
↓miR-130a	ATG2B	Enhanced apoptosis following starvation	[113]
↓miR-181a, b	Bcl-2, Mcl-1, XIAP, PTEN	Fludarabine sensitivity	[102,114]
↑miR-221	p27	Fludarabine resistance	[114]
** *Other targets* **			
↑miR-125b	CD20	Rituximab resistance	[115]
↑miR-532-3p	CD20	Rituximab resistance	[115]

Arrows (↓ and ↑) indicate miRNA downregulation or upregulation.

## Data Availability

Not applicable.

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
