# Peer review of "miRNAs in Lymphocytic Leukaemias—The miRror of Drug Resistance"

_ijms, 2022, doi:10.3390/ijms23094657_

Round 1

Reviewer 1 Report

This is a very comprehensive review of the literature on miRNAs in lymphocytic leukemias, especially with regard to their role in chemo resistance. This is a timely review of the subject and the authors have done a good job in framing the key pathways involved. 

Author Response

Dear Reviewer 1, 

We appreciate your comments. Some minor changes have been made to the manuscript and an additional figure (Figure 4) has been added to address the suggestions of Reviewer 2. 

Reviewer 2 Report

Yordan Sbirkov er al introduced the function and treatment strategies of miRNAs in lymphocytic leukemias, provided many details. But the logic of manuscript has some issues, here are my points:

Major:

  1. The structure of Part2/34 has problems. “Signaling pathways” and “Apoptosis/Autophagy” are not in a same category, they are not parallel. “Signaling pathways” indicates the molecular mechanism, but “Apoptosis” means a process of cell biology. Many signaling pathways can induce apoptosis.

Minor:

  1. Line 100, what is “cALL”?
  2. For Part2.1, I suggest the authors can summarize the signaling pathways in one picture.
  3. Line 337 “inluding”, a “c” is missed here.

Author Response

Dear Reviewer 2, 

We appreciate your comments and agree with the highlighted weakness and with the  suggested improvements. We have made an attempt to address these minor and major issues as follows:

Major:
1. We have tried to make a clearer distinction in the text between regulation of pathways (which may result in changes of different biological processes including apoptosis) and direct regulation of apoptosis. With the purpose to synopsize the plethora of different miRNA targets ( either involved in signaling pathways leading proliferation and cell survival or involved in processes like apoptosis and/or autophagy) we had already grouped these targets in 3 categories presented as separate subheadings. Now we have changed the titles of these subheadings hoping to improve the logic behind these 3 groups of miRNA targets and avoid confusion in the reader. Other examples of our efforts to clarify what is what can be found on lines: 
- 84-85
- 174-176
- 257-258
- 261
- 412
- 528-531
- 608-609

Minor:
1. "cALL" has been explained in line 53 and is hopefully clear from there on;
2. We thought the idea for another figure may be very useful to the readers. So we tried to summarise the pathways that are affected by miRNAs in all three malignancies discussed in this manuscript. In the end we focused on the two most frequently activated molecules - AKT and NF-kB and made a new figure -  Figure 4. 
3. The minor mistake on line 337 as well as other typos and such throughout the text have been corrected. 

We believe that with these changes, made thanks to Reviewer 2, the manuscript has considerably improved.   

Round 2

Reviewer 2 Report

The authors addressed my concerns.